# Dolutegravir Resistance in Mozambique: Insights from a Programmatic HIV Resistance Testing Intervention in a Highly Antiretroviral Therapy-Experienced Cohort

**DOI:** 10.3390/idr17050123

**Published:** 2025-09-30

**Authors:** Maria Ruano, Antonio Flores, Aleny Couto, Irénio Gaspar, Sabine Yerly, Ana Gabriela Gutierrez Zamudio, Rosa Bene, Adelina Maiela, Helder Macuacua, Jeff Lane, Florindo Mudender, Edy Nacarapa

**Affiliations:** 1I-TECH Mozambique “International Training & Education Center for Health”, Bairro Sommerschield, Avenue Cahora Bassa N# 106, Maputo P.O. Box 1102, Mozambique; mariar@itech-mozambique.org (M.R.); rbene@itech-mozambique.org (R.B.); helderm@itech-mozambique.org (H.M.); florindom@itech-mozambique.org (F.M.); 2Southern Africa Medical Unit, Médecins Sans Frontières, Cape Town P.O. Box 7925, South Africa; antonio.flores@joburg.msf.org; 3National STI/HIV/AIDS Program, Ministry of Health, Maputo P.O. Box 264, Mozambique; coutoaleny@gmail.com (A.C.); ireniogaspar@gmail.com (I.G.); 4Virology Laboratory, Hôpitaux Universitaires de Genève, P.O. Box 1205, 1211 Geneva, Switzerland; sabine.yerly@hcuge.ch; 5Médecins Sans Frontières, Maputo P.O. Box 1224, Mozambique; gabriela.gz@me.com; 6Department of Global Health, University of Washington, 908 Jefferson St., 12th Floor, Seattle, WA 98195, USA; lanej3@uw.edu

**Keywords:** HIV genotypic resistance testing, dolutegravir, HIV resistance, PLWH, Mozambique

## Abstract

**Background**: Treatment failure continues to play a role in HIV-related morbidity in Mozambique. Antiretroviral therapy (ART) regimen switches are decided empirically, as HIV genotypic resistance testing (HIV-GT) is unavailable in Mozambique’s public health system. Since 2016, Médecins Sans Frontières (MSF) and I-TECH have provided access to HIV-GT at Alto Maé Health Center, Maputo. We describe the cohort of people with virologic failure (VF) that underwent HIV-GT and analyze dolutegravir (DTG) resistance (R) patterns. **Methods**: This cross-sectional assessment of routine programmatic data between July 2020 and February 2024 was conducted to guide future program enhancements. People living with HIV (PLWH) receiving ART beyond the first line with confirmed VF were included. Mutations were interpreted according to the Stanford HIVdb algorithm. We applied Bayesian bootstrapping for analysis, and the threshold for significance of effects was defined as a probability of 95%. **Results**: A total of 106 persons underwent HIV-GT following a structured adherence strategy, 62 (58.5%) of whom were on a DTG-based regimen. Fifty-seven of the 62 samples from persons on a DTG-based regimen were sequenced, and 51 (89.5% [95% CrI: 80.7, 96.2]) had confirmed resistance to DTG; the mean DTG-R score was 70.2 (95% CrI: 62.2, 78). Samples with DTG-R had a median of three INSTI mutations (IQR 1–4). Major DTG-associated mutations were found in 46 out of 57 samples: G118R (*n* = 28), R263K (*n* = 15), and Q148RK (*n* = 7). None of the people on the protease inhibitor regimen had an INSTI mutation. **Conclusions**: In contexts with limited access to resistance testing, the introduction of algorithms to identify PLWH at risk of developing drug resistance is strongly recommended. The proposed algorithm incorporates adherence reinforcement strategies, as recommended in national policies, followed by a short, supervised antiretroviral therapy (ART) support strategy. This approach has shown a high predictive value for identifying PLWH with resistance mutations to dolutegravir (DTG), thereby allowing the continuation of the effective DTG regimen without unnecessary regimen switches.

## 1. Introduction

Millions of individuals in Africa have been receiving antiretroviral treatment (ART) based on dolutegravir (DTG) since the significant transition to new treatment regimens in 2019–2020. This shift was primarily driven by the World Health Organization (WHO), which advocated for DTG as the preferred option for both first- and second-line ART due to its substantial clinical benefits in terms of efficacy, high barrier to resistance, and favorable safety profile [1]. However, limited guidance has been provided to countries on identifying and managing resistance to DTG in scenarios with restricted access to resistance testing, such as in Mozambique. Many countries continue to encounter significant barriers to implementing new algorithms that incorporate resistance testing capabilities [2].

Dolutegravir remains a transformative tool in Mozambique’s HIV response, enabling 95% viral suppression rates in optimized programs. However, its safety advantages are compromised by systemic gaps in adherence support and resistance monitoring [3]. Urgent scale-up of viral load testing, targeted adherence interventions for high-risk groups, and sentinel resistance surveillance are essential to prevent widespread resistance [4]. The experience underscores that even high-barrier drugs like DTG are vulnerable to suboptimal implementation, particularly in regions dominated by HIV subtype C [5].

In Mozambique, the decision to switch ART regimens is made based on pragmatic clinical observations due to the unavailability of HIV genotypic resistance testing (HIV-GT) within the public health system [6]. With the advent of DTG and the increasing availability of more consistent treatment regimens, alongside expanding access to viral load monitoring, suppression rates have improved significantly [1,7]. Consequently, only a small percentage of PLWH (people living with HIV) may require a change in their treatment regimen [1].

In this scenario, it is essential to establish innovative strategies for identifying PLWH who require resistance testing and, ultimately, a revised ART regimen.

Since 2016, efforts by Médecins Sans Frontières (MSF) and, subsequently, the International Training and Education Center for Health (I-TECH-Mozambique [8]) have facilitated access to HIV-GT at the Alto Maé Referral Center in Maputo. This report presents data from a programmatic intervention focused on HIV genotype resistance testing within a cohort of PLWH with extensive exposure to ART in a resource-limited setting. The aim of this analysis is to share our field experience in identifying cost-effective strategies to detect ART resistance in a resource-limited setting.

This evaluation aims to assess the ability to detect resistance within our patient flow. Patient management is individualized based on genotype test results. Most cases exhibiting dolutegravir (DTG) resistance were offered salvage therapy consisting of a boosted protease inhibitor (PI) and two nucleoside reverse transcriptase inhibitors (NRTIs) [9]. However, in specific instances where DTG retained partial activity—particularly when extensive resistance to NRTIs was present—DTG was maintained within the treatment regimen.

## 2. Materials and Methods

### 2.1. Setting

According to the 2021 national survey on the impact of HIV in Mozambique (INSIDA), it is estimated that the prevalence of HIV in Maputo city stands at 16.2% among individuals aged 15 and older [10]. Furthermore, as reported by the Health Information System for Monitoring and Evaluation (SIS-MA) of the MoH (Ministry of Health), Maputo city had a cumulative total of 170,161 people living with HIV (PLWH) who were receiving antiretroviral therapy (ART) by mid-2024 [11].

In 2003, Médecins Sans Frontières (MSF) established the Centro de Referência de Alto-Mae (CRAM) in collaboration with the Ministry of Health (MoH) as an outpatient center dedicated to the referral of PLWH with advanced HIV disease (AHD) and those suspected of ART failure and in need of second-line ART regimens from the broader Maputo urban area [8]. In December 2020, management of CRAM was transitioned to ITECH-Mozambique [12]. The center continues to receive PLWH from the health network within Maputo city.

The CRAM database maintains a cumulative registry of 25,432 individuals living with HIV, of which 1819 remain actively engaged in care as of July 2024. CRAM provides a range of outpatient services, including rapid screening and management of AHD, chemotherapy for Kaposi Sarcoma, and treatment of multidrug resistant tuberculosis, as well as confirmation and follow-up care for PLWH exhibiting resistance to ART [13].

### 2.2. Study Design and Population

This cross-sectional assessment of routine programmatic data was conducted to guide future program enhancements. The included data were retrospectively collected as part of the treatment failure program (TFP) from CRAM between July 2020 and February 2024. Participants selected for HIV drug-resistance (HIVDR) testing were included.

#### 2.2.1. Eligible Participants

The eligibility criteria for accessing a resistance test at CRAM, as per our internal algorithm, included the following: first, non-naive PLWH on ART, irrespective of WHO/AIDS clinical staging and/or CD4 count; second, PLWH who had been exposed to ART for a minimum of 24 months; third, individuals exhibiting persistent high viral load, defined as at least two consecutive results exceeding 1000 copies/mL, prior to enrollment in the TFP; fourth, persistent high viral load following at least three consecutive adherence counseling sessions after entering the treatment failure program; fifth, persistently high viral load after a 4-week supervised treatment intervention.

The sole exclusion criterion for eligibility to undergo a resistance test was a viral load result of less than 5000 copies/mL after adherence support interventions, as per the laboratory standard operating practice. This limitation arises from the inherent constraints of dried blood spot (DBS) samples, which are not capable of effectively amplifying and sequencing viral genetic material at lower viral load levels [14].

##### Algorithm for Selecting PLWH for Resistance Testing

PLWH with persistent high viral loads were referred to CRAM and were managed according to the outlined algorithmic steps (Figure 1):

Once PLWH were selected for this intervention, monthly adherence counseling sessions were conducted by senior lay counselors, coinciding with clinical consultations by a physician. A minimum of three sessions were provided prior to viral load (VL) re-evaluation. This is the current national recommendation for any individual on non-suppressive ART. Additionally, if VL remained elevated, a four-week supervised antiretroviral therapy (ART) period was initiated. This involved daily communication via phone call or text message with the patient to confirm adherence to the treatment. During this period, PLWH were required to report daily on their treatment compliance. Following this intervention, all PLWH provided a blood sample for two purposes: to conduct a new VL test and to collect a dried blood spot sample on filter paper for storage. A resistance test was performed if VL remained elevated, typically when it exceeded 5000 copies/mL.

##### Positive Predictive Value (PPV)

The positive predictive value (PPV) is defined as the proportion of individuals with the target condition (i.e., those who tested positive for resistance to a certain drug) among all those who tested positive according to the diagnostic algorithm. Mathematically, PPV is calculated as the ratio of true positives (TPs) to the sum of true positives and false positives (FPs).

##### Genotypic HIV Drug-Resistance (HIVDR) Testing

Four milliliters of whole blood samples were collected by venipuncture in a K2 EDTA tube. Resistance genotyping was performed on dried blood spot (DBS) samples using 50 μL of venous blood per spot and dried overnight; samples were stored at −20 °C for less than 1 month before being shipped to the virology laboratory (Hôpitaux Universitaires de Genève) [15].

The DBS samples were cut and subjected to elution for the release of viral RNA. Reverse transcription, amplification, and Sanger sequencing were performed as previously described [16,17,18].

The Stanford HIVdb algorithm was used to interpret the results of the genotypic tests [10]. The numbers of drug class-associated mutations and penalty (resistance [R]) scores for tenofovir (TDF), zidovudine (AZT), atazanavir (ATV), and dolutegravir (DTG) are reported [19].

Resistance (of any level: from potential low level to high level) was defined as a penalty score ≥ 5.

We defined *major* dolutegravir resistance (DTG-R) mutations as those that can cause at least intermediate resistance to DTG when they occur in isolation [19]. These mutations are G118R, Q148K/R, and R263K for the purposes of this analysis. *Non-major* mutations are all other mutations (i.e., accessory or major mutations related to INSTIs but which do not cause resistance or cause only low-level resistance to DTG when they arise alone) [19].

#### 2.2.2. Sampling and Sample Size of Program Participants

The sample encompassed the entire population of non-naive PLWH on ART who experienced VF and had undergone HIV-1 resistance testing, as recorded in the HIV-1 resistance test registration book at CRAM, in accordance with the specified eligibility criteria. As this was a convenience sample from a real-world setting, an a priori sample size calculation was not applicable.

#### 2.2.3. Program Data Collection

The study team extracted routine clinical data from paper-based patient files during the period under analysis, between the years 2020 and 2024. Exposure variables were categorized into six fields: (1) the date of ART initiation and HIV-GT collection; (2) demographic profile: gender and age; (3) the ART drug combination at the time of the genotype sample collection: NRTIs-nucleotide reverse transcriptase inhibitors (ABC-Abacavir, AZT-Zidovudine, TDF-Tenofovir) plus PIs-protease inhibitors (LPV-Lopinavir, RTV-Ritonavir, ATV-Atazanavir, DRV-Darunavir); or NRTI plus INSTI-integrase strand transfer inhibitor (DTG-Dolutegravir); (4) previous and most recent viral load; (5) major HIV-1 drug-resistance mutations described by their categories: NRTI, PI, and INSTI; (6) most recent CD4 T lymphocyte count (measured using PIMA technology).

Data collected from eligible PLWH were anonymized: each patient was given an alphanumeric code, and the anonymized data were included in the datasheet. Spreadsheet data were exported to R for further data analysis.

#### 2.2.4. Program Data Analysis

All analyses were carried out in *R* (version 4.4.1). Bayesian bootstrap implementation was performed using the bayesboot package (version 0.2.2).

We used medians and interquartile ranges (IQRs) to describe continuous or count (discrete) variables and proportions (%) to summarize categorical data. Our choice of Bayesian methods over frequentist approaches is motivated by their ability to quantify parameter uncertainty, particularly under smaller sample sizes reflective of real-world programmatic conditions. To further avoid parametric assumptions, we employed nonparametric bootstrapping for distribution assessment. The underlying assumption is that this cohort represents the population of PLWH with virological failure in the treatment failure program at CRAM.

To report uncertainty, we used the highest density interval for the 95% Bayesian credible interval (95% CrI). For the prior distribution of weights in the bootstrap algorithm, a non-informative uniform Dirichlet distribution was used. For proportions, a non-informative Jeffreys prior Beta (0.5, 0.5) was applied.

Also, to determine the existence of an effect, we calculated the probability of direction (*pd*), which is the probability that the posterior distribution goes in the direction of the mean effect (positive or negative for differences or above or below 1 for ratios). Regarding the existence of effect, the *pd* was interpreted as follows: *pd* ≤ 95%, uncertain effect; 95% < *pd* ≤ 97.5%, possibly existing; 97.5% < *pd* ≤ 99%, likely existing; *pd* > 99%, probably/certainly existing.

## 3. Results

### 3.1. Cohort Main Characteristics

Among the 106 samples successfully sequenced, 99 (93.4%) were subtype C; 62 (58.5%) were from individuals on a DTG-based regimen; 57 (53.8%) were female; 80 (75.5%) were receiving a TDF-based backbone; and 72 (67.9%) had advanced HIV disease (defined as a CD4 count equal to or lower than 200 cells/mm^3^) (Table 1).

All cases had confirmed virological failure (persistently high viral load). The average time on ART for the cohort was 11.6 years (IQR 9.1–14.4). The average time on the current ART regimen was 3.1 years (IQR 2.2–4.5) (Table 1). A difference in immunological status was observed between female and male PLWH in this cohort. Men exhibited a lower mean CD4 count and a higher proportion of advanced HIV disease, with high probability in both instances (prevalence difference 24.6%; 95% CrI 2.2–46) (Figure 2).

### 3.2. INSTI Mutations and Resistance

Fifty-seven (92%) of the 62 samples from PLWH on a DTG-based regimen were sequenced, and 51 (89.5% [95% CrI: 80.7, 96.2]) had confirmed resistance to DTG; the mean DTG-R score was 70.2 (95% CrI: 62.2, 78). PLWH with DTG-R had a median of three INSTI mutations (IQR 1–4) (Table 2).

Major DTG-associated mutations were found in 46 out of 57 samples (80.7%): G118R (n = 28), R263K (n = 15), and Q148RK (n = 7). However, the most frequent mutation found was E138K (n = 33), a non-major mutation that only confers potential low-level resistance when present in isolation (Figure 3).

We found a high resistance level to NRTIs in our cohort. The median NRTI mutations in the global cohort were three (IQR 2–5), with a median of four (2–5) mutations in the DTG group compared to three (2–5) mutations in the PI group (Table 2).

We did not find any INSTI resistance mutations in PLWH in the PI group (n = 44). None of them had received treatment with INSTIs before (Table 2).

## 4. Discussion

### 4.1. Main Findings

In this programmatic HIV-GT intervention, DTG-R was very frequently found in highly ART-exposed persons on a DTG-based ART regimen experiencing VF after going through a structured adherence algorithm that included a short, supervised treatment intervention.

In our cohort, 51 cases of DTG resistance were detected among 57 selected PLWH, resulting in a resistance ratio of 89.5% in individuals experiencing confirmed virological failure. This finding indicates high resistance detection within our cohort.

DTG resistance detection surpassed the 19.6% (36/183) reported in a 2021–2022 Mozambican clinic-based study of experienced individuals transitioning to DTG across seven high-volume sites [19]. Our analysis found higher DTG resistance than the 26.9% (24/89) reported in a Malawian retrospective cohort [20], and it likewise exceeded the 5.8% (4/84) INSTI resistance observed in Tanzania’s 2020 national HIV drug-resistance survey [21]. The differences observed in our study result from the analysis of a highly filtered cohort, selected using an algorithm designed to limit resistance testing in patients who—with additional adherence support—were likely to achieve viral suppression.

The results of this tightly controlled cohort of PLWH underscore the critical need for the development and implementation of high-quality, evidence-based adherence interventions, complemented by access to drug resistance testing for effective clinical management, particularly in the subgroup of individuals heavily exposed to ART.

As this study relied on real-world programmatic data, detailed historical viral load testing records were often incomplete or unavailable for analysis. However, available programmatic evidence indicates substantial delays in initial high viral load detection, frequently exceeding 6 months, due to gaps in routine monitoring and reporting. When combined with the standard 4-month enhanced adherence counseling (EAC) process prior to resistance testing, our conservative estimate suggests that participants in the pre-resistance testing ART support algorithm experienced at least 11 months of viremia before regimen adjustment.

Our algorithm demonstrated a notably high positive predictive value for finding resistance among individuals with longstanding exposure to ART. The analysis demonstrated that approximately 89% of those identified as “algorithm-positive” (i.e., non-suppressors despite adherence interventions) had a confirmed positive resistance test, irrespective of the resistance level. Thus, the PPV of the algorithm for predicting resistance was 89%.

It is plausible to consider that the elevated prevalence of antiretroviral resistance observed in this cohort may have been partially influenced by delays in case identification, potentially associated with the diagnostic algorithm employed. In clinical practice, this patient cohort remained in virological failure for a prolonged period prior to referral to our healthcare unit, which may have contributed to the accumulation of resistance mutations.

In healthcare settings with broader access to genotypic resistance testing, reliance on such algorithms might be less critical, as earlier detection could facilitate more timely therapeutic adjustments and potentially mitigate the development of resistance. Nevertheless, in our context, this approach allowed the strategic allocation of resistance testing resources to PLWH with increased probability of carrying resistance mutations, thus enabling customized decision-making concerning the subsequent individualized treatment options.

When comparing various mutational patterns identified in previous cohort analyses, our study confirmed a subtype C predominance and the distribution of its resistance mutations across subtypes (A, B, C, and U). This variation aligns with known differences in mutational patterns between HIV subtypes (A, C, D, URFs) in Southern and Eastern Africa, a region predominantly affected by subtype C [5]. The RESIST study, which evaluated the genotypic resistance test results of PLWH undergoing DTG regimens from 2013 to 2021 (n = 599; INSTI drug-resistance mutations detected in 86 PLWH, representing 14%), revealed a lower level of resistance alongside a distinct mutational pattern. The most prevalent major DTG-associated mutation was R263K, identified in 10 out of 36 samples (28%). The second most frequent DTG-associated mutation was Q138K, found in 7 out of 36 samples (19%), while the third most common major mutation was E148K, present in 6 out of 36 samples (17%). Notably, only 3 out of 36 PLWH (8%) exhibited the G118K mutation, which is recognized as the major mutation with the most significant impact on susceptibility to dolutegravir [22].

In our cohort, 67% of samples had high-level DTG resistance (Stanford score ≥ 60) compared to 17% in the RESIST study. Conversely, 6% of our samples had low-level resistance (Stanford score < 15) versus 36% in the RESIST study [22]. The observed differences may reflect varying stages in resistance development. While the RESIST study cohort, largely from settings with prompt resistance testing access, included only nine (2%) DTG-resistant samples from Africa [22], the CRAM cohort had prolonged exposure to suboptimal DTG regimens prior to testing. This is likely to explain the higher resistance levels and greater detection of major DTG mutations in our cohort.

In the cohort of PLWH on a protease inhibitor-based regimen, there was no recorded prior exposure to dolutegravir (DTG), and this allows us to affirm that no primary resistance to integrase strand transfer inhibitors (INSTIs) was identified.

### 4.2. Other Relevant Findings

Extended resistance to other ART classes was observed, showing a high detection of NRTI resistance in the cohort (overall, 76.5% for TDF and 71.5% for AZT), including in the DTG subgroup.

Men in our cohort had lower mean CD4 counts and a higher proportion of advanced HIV disease (AHD) than women. This aligns with findings that 31% of men starting ART in Mozambique had AHD compared to 21% of women [15]. CRAM’s focus on AHD and ART failure may explain the higher male representation, contrasting with the feminized HIV epidemic in sub-Saharan Africa, where women comprise over 57% of PLWH [16]. Other reports also indicate higher AHD prevalence in men [23].

### 4.3. Limitations of the Study

Due to the necessity of transporting materials outside of Mozambique, we used DBS technology to facilitate the collection, preservation, and transport of samples. While this technology offers certain advantages, it also presents limitations, particularly in its sensitivity to detecting mutations in samples exhibiting lower-level viremia. Consequently, PLWH with a viral load below 5000 copies/mL following the intervention were typically excluded from resistance testing. Utilizing plasma samples would likely have allowed for the inclusion of a greater number of PLWH in the testing strategy, thereby enhancing its overall success.

Due to the clinical focus of this intervention, designed to enhance tailored decision-making, exceptions allowed resistance testing for three PLWH with viral load results below the standard threshold (1750, 3410, and 4800 copies/mL). Only the sample with 4800 copies/mL was successfully amplified and sequenced, highlighting challenges with genotyping DBS in lower-level viremia and reduced amplification success. By requiring a minimum of 24 months of ART exposure, we may have excluded some PLWH deemed unlikely to develop resistance. Most unsuppressed PLWH on first-line DTG achieved viral suppression through adherence reinforcement standardized in national protocols.

Additionally, the cross-sectional nature of this analysis did not allow for the evaluation of ART management (regimen switch) based on resistance test results, limiting the understanding of the impact of HIV resistance testing on longer-term outcomes. This reflects the programmatic focus of the intervention, which prioritized strategic resource allocation for genotyping over longitudinal follow-up of treatment outcomes.

## 5. Conclusions and Recommendations

In a context with scarce access to resistance testing capacity, we strongly recommend the validation and introduction of algorithms designed to identify PLWH at high risk of developing resistance. Our algorithm includes adherence reinforcement strategies already recommended in national policies, followed by an additional short, supervised ART support strategy. This additional step demonstrated a very high predictive value for identifying PLWH with DTG-associated resistance mutations. Importantly, this allows for the avoidance of unnecessary regimen switches, i.e., in PLWH for whom the highly effective DTG regimen can still achieve viral suppression.

Because the positive predictive value of a test to identify a particular event (in this case, the existence of resistance) is partly related to the prevalence of this event, it is uncertain how this algorithm may perform in PLWH for whom resistance is less probable, as is the case of those using DTG as a first line or with a shorter time of exposure to ART. Further assessment of this algorithm in distinct subpopulations is warranted. If feasible, plasma-based testing should be considered to improve detection of resistance at lower VL cut-offs.

### What Are the Bullet Points of This Study?

**High Detection of Dolutegravir Resistance**: The study reveals a strikingly high detection of dolutegravir (DTG) resistance (89.5%) among individuals with virologic failure (VF) who had extensive prior exposure to antiretroviral therapy (ART). This underscores the risk of resistance in heavily ART-experienced populations and challenges the assumption that DTG resistance is still rare.**Effective Algorithm for Resistance Identification**: The study outlines a structured algorithm combining viral load monitoring, adherence reinforcement, and supervised ART support to identify individuals at high risk of resistance. The add-on approach demonstrated a high positive predictive value (89.5%), enabling targeted use of genotypic resistance testing where resources are limited.**Mutational Patterns and Resistance Scores**: The study documents the most common DTG-associated mutations (e.g., G118R, R263K, Q148RK) and provides detailed resistance scores, offering insights into the genetic basis of DTG resistance in this cohort.**Programmatic Implications**: The findings highlight the need for integrating resistance testing into ART programs in resource-limited settings, particularly for individuals with prolonged ART exposure and virologic failure. The study substantiates the need for adherence support and supervised interventions before considering regimen switches, which can help preserve effective treatment options like DTG. It also highlights the importance of elucidating the role of regular HIVDR surveillance vs. clinically driven individual resistance testing in settings where the affordability and feasibility of HIVDR testing are perceived as limited.**Sex Disparities in Advanced HIV Disease**: The study notes significant immunological differences between men and women in the cohort, with men exhibiting lower CD4 counts and higher rates of advanced HIV disease. This aligns with broader trends in sub-Saharan Africa and emphasizes the need for targeted interventions for male populations.**Limitations and Future Directions**: The study acknowledges the challenges of using dried blood spot (DBS) samples for resistance testing, particularly their lower sensitivity for detecting resistance in cases of low-level viremia. It calls for further research to validate the algorithm in different subpopulations and to explore plasma-based testing for improved sensitivity.

## Figures and Tables

**Figure 1 idr-17-00123-f001:**
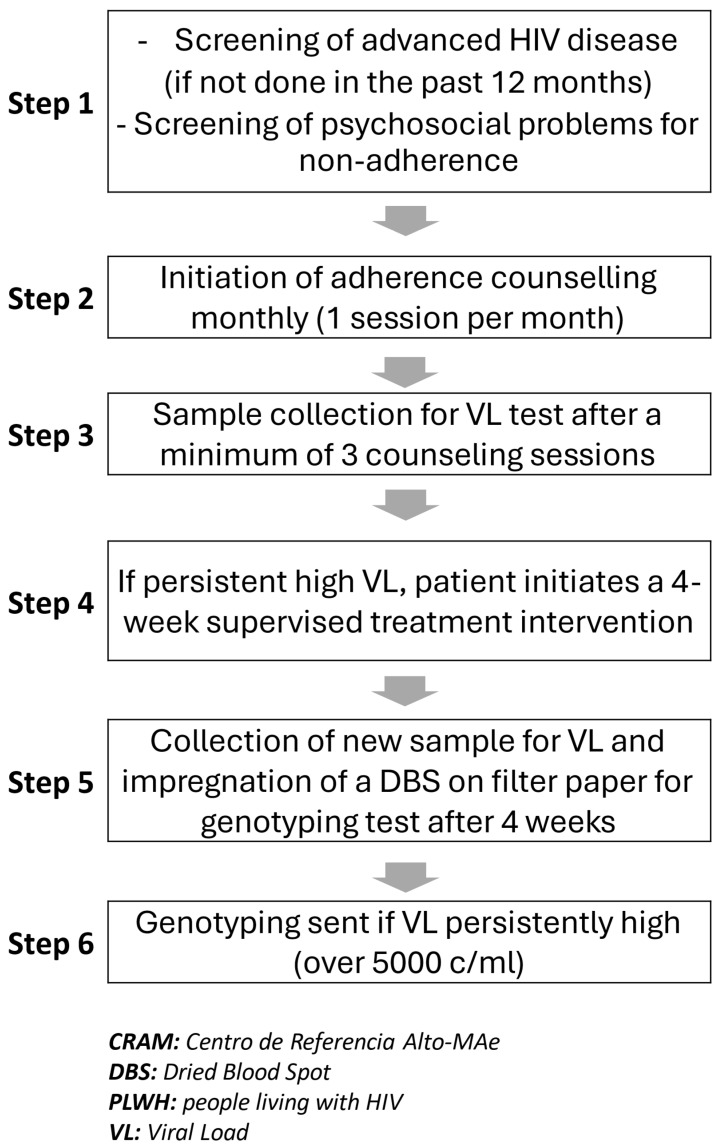
Algorithm for selection of PLWH submitted to a genotyping test at CRAM.

**Figure 2 idr-17-00123-f002:**
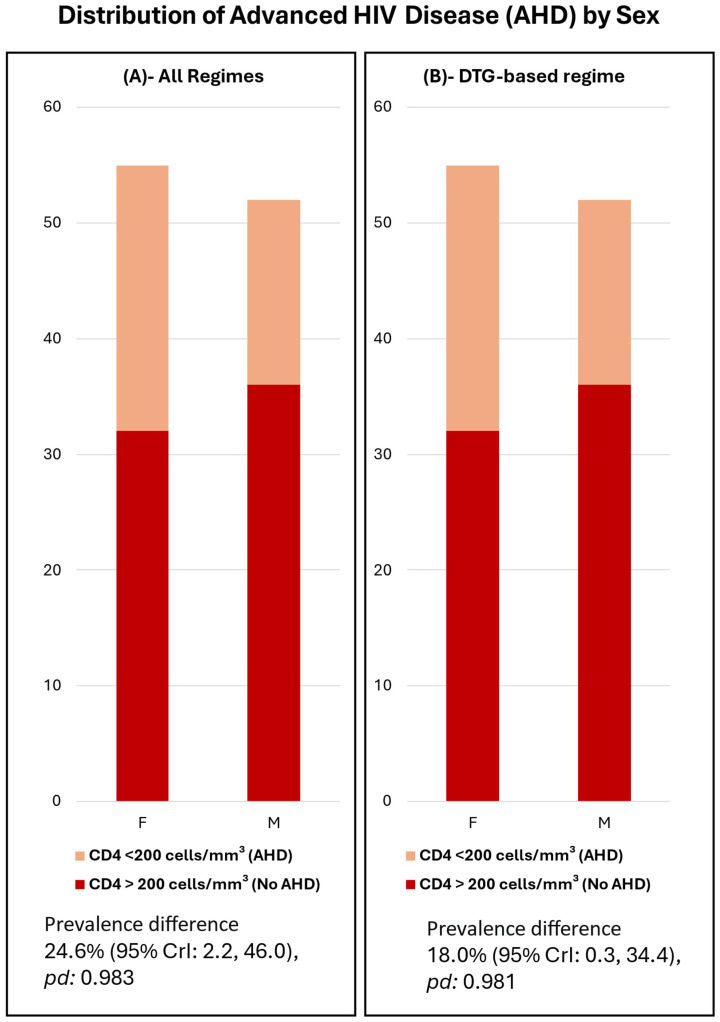
(**A**,**B**) illustrated a high overall burden of advanced HIV immunosuppression, with a significant proportion of ART patients presenting with CD4 counts ≤ 200 cells/mm^3^. The analysis revealed a major sex disparity, demonstrating that males were significantly more likely to present with Advanced HIV Disease compared to females. This disparity was quantified as a prevalence difference of +24.6% across all treatment regimens (**A**) and +18.0% specifically among patients on DTG-based regimens (**B**).

**Figure 3 idr-17-00123-f003:**
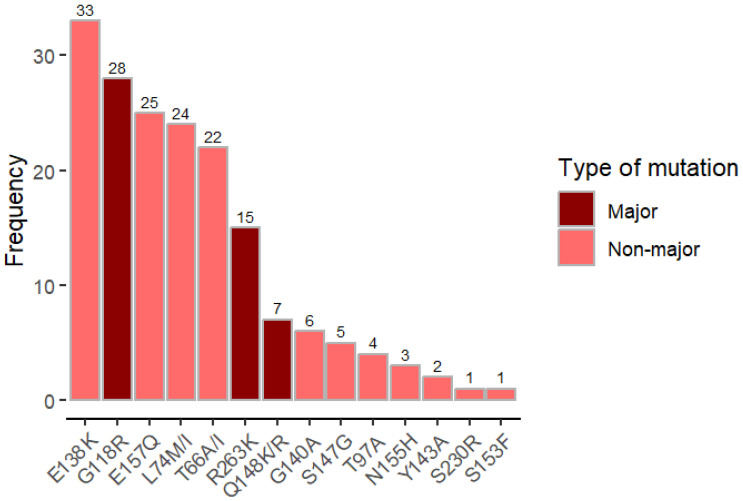
Frequency of DTG mutations among individuals on DTG and with DTG-resistance mutations. This figure shows that major dolutegravir (DTG)-associated resistance mutations were detected in 46 of 57 samples (80.7%). The specific major mutations identified were G118R (28 samples), R263K (15 samples), and Q148R/K (7 samples). However, the most frequently observed mutation overall was E138K (33 samples), which is not classified as a major DTG mutation and typically confers only low-level resistance when it occurs in isolation.

**Table 1 idr-17-00123-t001:** Cohort main characteristics.

	All	DTG-Based	PI-Based
N	106	62	44
Sex Women, N (%)	57 (53.8)	30 (48.4)	27 (61.4)
Men, N (%)	49 (46.2)	35 (53.8)	17 (38.6)
Age, median (IQR)	42 (38, 48)	42 (38, 48)	42 (37, 50)
Time on ART (years), median (IQR)	11.6 (9.1, 14.5)	11.2 (6.5, 14.4)	12.1 (10.4, 14.6)
Time on current ART regimen (years), median (IQR)	3.2 (2.2, 4.5)	2.5 (1.9, 3.3)	4.8 (3.7, 6.0)
TDF backbone, N (%)	80 (75.5)	49 (79.0)	31 (70.5)
Non-first-line ART, N (%)	102 (96.2)	60 (96.8)	42 (95.5)
Viral load (log10), median (IQR)	4.6 (4.2, 5.2)	4.7 (4.3, 5.2)	4.6 (4.2, 4.9)
CD4, median (IQR)	134 (58, 234)	128 (56, 233)	146 (59, 258)
Advanced HIV, N (%) ^1^	72 (67.9)	43 (69.4)	29 (65.9)
Subtype C, N (%)	99 (93.4)	58 (93.5)	41 (93.2)

DTG, dolutegravir; TDF, tenofovir; IQR, interquartile range; PI, protease inhibitor; INSTI, integrase strand transfer inhibitor. ^1^ Advanced HIV was defined as CD4 count equal to or below 200 cells per mm^3^.

**Table 2 idr-17-00123-t002:** Resistance profile of PLWH.

	All	DTG-Based	PI-Based
Number of NRTI mutations, median (IQR) ^1^	3 (2, 5)	4 (2, 5)	3 (2, 5)
TDF-R score, median (IQR)	20 (5, 35)	20 (13.8, 36.2)	15 (0, 28.8)
TDF-R score ≥ 5, N (%)	78 (76.5)	48 (80.0)	30 (71.4)
AZT-R score, median (IQR)	57.5 (0, 90)	60 (0, 90)	55 (5, 88.8)
AZT-R score ≥ 5, N (%)	73 (71.5)	41 (68.3)	32 (76.2)
Number of PI mutations, median (IQR) ^2^	2 (0, 5)	1 (0, 2)	5 (4, 7)
ATV-R score, median (IQR)	0 (0, 70)	0 (0, 0)	65 (17.5, 110)
ATV-R score ≥ 5, N (%)	51 (49.5)	12 (20.0)	39 (90.7)
DRV-R score, median (IQR)	0 (0, 0)	0 (0, 0)	0 (0, 0)
DRV-R score ≥ 5, N (%)	14 (13.6)	6 (10.0)	8 (18.6)
Number of INSTI mutations, median (IQR) ^3^	0.5 (0, 3)	3 (1, 4)	0 (0, 0)
DTG-R score, median (IQR)	30 (0, 80)	75 (40, 81.2)	0 (0, 0)
DTG-R score ≥ 5, N (%)	51 (53.6%)	51 (91.1%)	0 (0%)

ATV, atazanavir; AZT, zidovudine; DRV, darunavir; DTG, dolutegravir; TDF, tenofovir; R, resistance. IQR, interquartile range; PI, protease inhibitor; INSTI, integrase strand transfer inhibitor. ^1^ Reverse transcriptase not amplified, N (%): all, 4 (3.7%); DTG, 2 (3.2%); PI, 2 (4.4%). ^2^ Protease not amplified, N (%): all, 3 (2.8%); DTG, 2 (3.2%); PI, 1 (2.3%). ^3^ Integrase not amplified, N (%): all, 11 (10.3%); DTG, 6 (9.7%); PI, 5 (11.4%).

## Data Availability

The datasets utilized in this study are available from the corresponding author upon reasonable request; however, they are not publicly accessible due to privacy constraints.

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
