# Peer review of "Dolutegravir Resistance in Mozambique: Insights from a Programmatic HIV Resistance Testing Intervention in a Highly Antiretroviral Therapy-Experienced Cohort"

_2036-7449, 2025, doi:10.3390/idr17050123_

Round 1

Reviewer 1 Report (Previous Reviewer 1)

Comments and Suggestions for Authors

The authors have addressed all the points, though not all fully satisfactory. In addition, there are some new points to pay attention to.

The manuscript is fairly well written, though could benefit from textual review by a native English speaker.

General major points

  1. Given the title, what is the relevance of including results of individuals on PI-based regimens in this manuscript? This goes back to the main aim of the study, that is still not outlined clearly enough at the end of the Background
  2. The concept of a relative DTG resistance score as an indicator is unconventional and I have doubts about its relevance and validity. If I understand correctly it wants to provide a score of the impact of resistance relative to the number of specific mutations. First, why would this be important? Second, the number of mutations to a drug is not decisive, as there are very large differences in the impact of various DRMs on resistance, viral replication and viral fitness. Third, there are synergistic effects between some mutations and not between others. Fourth, some mutations are nearly always only seen when initial (major) mutations have appeared, i.e. compensatory mutations that increase fitness. These elements do not seem to be captured by the concept of a relative DTG resistance score. In the Discussion, the authors try to justify the inclusion of this score, in lines 389-397, but the suggestions are speculative and do not appear to be founded on evidence. Unless the authors have a good reference to this indicator in the literature, I recommend leaving this concept out.
  3. Line 354. “This approach allowed the strategic allocation of resistance testing resources to PLWH with increased probability of carrying resistance mutations enabling customized decision-making concerning the subsequent individualized treatment options.” This may be so, but given the very high percentage of high-level resistance and the very high percentage of AHD, it could just as well be argued that the virological failure management strategy failed these individuals and that the protracted diagnostic MSF-iTECH algorithm contributed to this. It would be important that the authors acknowledge this. This does not disqualify their findings, but adds to the argument for better, evidence-based management of virological failure on DTG-based regimens in this setting (such evidence is currently not available) that may include wider availability of HIVDR testing and other approaches. In my view this limitation of the algorithm used in the study should come out more clearly in the Discussion.
  4. I would have expected that the authors would advocate for the ability to do in-country HIVDR testing for regular HIVDR surveillance and possibly for individual case management (vs. sending samples to another continent).

Specific minor points

Line 71. “Mozambique's lessons offer critical insights for other sub-Saharan African countries navigating DTG transitions”. This is not a sentence that fits into a Background section, it would be expected in the Discussion, and the only of course if “critical lessons” can be learned from  the findings.

Line 121. “Eligible participants included individuals of all ages who had access to HIV-GT.” I think the wording is not accurate. Participants were included if they were selected for HIVDR testing.

Line 158. “those who tested positive for resistance” – indicate if this is DTG resistance, or if not, clarify what kind of resistance is meant

Line 207. “ART drug combination”. Is this the regimen that the individual was on when the sample for sequencing was taken? please clarify.

Line 246. “Subtype C 99 (93.4%)” This is a repeat. Please remove and in addition, please decide on the number of decimals for percentages and numbers and use this consistently throughout text, tables and figures

Line 249. It is argued that men have a higher percentage of AHD than women but neither in the text, nor in Figure 2 are point estimates provided, which makes it hard to gauge the clinical relevance and magnitude of this finding. Please provide point estimates and numbers of the column totals and categories

Line 277. “so we assume we didn´t find primary resistance”. This is a conclusion that should appear in the Discussion, not the Results section

Line 278. PI resistance mutations were seen among individuals on a DTG regimen, had they been exposed to PIs in previous regimens? Such information would be valuable

Line 321. “resistance rate”. Given that percentages are presented, I would suggest saying ratio instead of rate

Lines 355-374. This paragraph contains much repetition of results and the relevance of the findings is not made very clear. I suggest considerable shortening and providing meaning of the mutation pattern findings observed (if there is any).

Author Response

August 31, 2025

Infectious Disease reports (idr),

EBM - Editorial Board Member

  • Steve Chen
  • Marija Jovanović

Dear Editors:

Please find attached a revised version of our manuscript, “Dolutegravir resistance in Mozambique:  insights from a programmatic HIV resistance testing intervention in a highly antiretroviral therapy experienced cohort  ,” (Manuscript idr-3726835 or idr-3618516). We have modified the original submission taking into consideration all comments and suggestions made by you. All changes have been highlighted in tracked changes on revised document. Below, please find an itemized response.

Answer (A): We appreciate your thanks and, considering it extremely important for the enrichment of the article, we added, deleted, rewrote, and reorganized the recommended points, according to your suggestions throughout the document.

Reviewer 1

The authors have addressed all the points, though not all fully satisfactory. In addition, there are some new points to pay attention to.

The manuscript is fairly well written, though could benefit from textual review by a native English speaker.

General major points

  1. Given the title, what is the relevance of including results of individuals on PI-based regimens in this manuscript? This goes back to the main aim of the study, that is still not outlined clearly enough at the end of the Background

Response:

This article brings the results of a real-world cohort in this context. We focus on the DTG sub-cohort due to the relevance of the findings and the importance of the INSTI Class.

The fact that no INSTI resistance mutations were found in the IP group seems relevant to us, as we can say that we are not finding primary resistance mutations in a real-world population.

  1. The concept of a relative DTG resistance score as an indicator is unconventional and I have doubts about its relevance and validity. If I understand correctly it wants to provide a score of the impact of resistance relative to the number of specific mutations. First, why would this be important? Second, the number of mutations to a drug is not decisive, as there are very large differences in the impact of various DRMs on resistance, viral replication and viral fitness. Third, there are synergistic effects between some mutations and not between others. Fourth, some mutations are nearly always only seen when initial (major) mutations have appeared, i.e. compensatory mutations that increase fitness. These elements do not seem to be captured by the concept of a relative DTG resistance score. In the Discussion, the authors try to justify the inclusion of this score, in lines 389-397, but the suggestions are speculative and do not appear to be founded on evidence. Unless the authors have a good reference to this indicator in the literature, I recommend leaving this concept out.

Response:

We agree to remove this way of referring to resistance (relative DTG resistance score)

  1. Line 354. “This approach allowed the strategic allocation of resistance testing resources to PLWH with increased probability of carrying resistance mutations enabling customized decision-making concerning the subsequent individualized treatment options.” This may be so, but given the very high percentage of high-level resistance and the very high percentage of AHD, it could just as well be argued that the virological failure management strategy failed these individuals and that the protracted diagnostic MSF-iTECH algorithm contributed to this. It would be important that the authors acknowledge this. This does not disqualify their findings, but adds to the argument for better, evidence-based management of virological failure on DTG-based regimens in this setting (such evidence is currently not available) that may include wider availability of HIVDR testing and other approaches. In my view this limitation of the algorithm used in the study should come out more clearly in the Discussion.

Response:

Undoubtedly, limited access to genotype testing is a key factor contributing to the continued use of potentially ineffective treatment regimens. A strategic approach to deploying this scarce resource—by offering it only to patients who fail to achieve viral suppression after a brief, supervised treatment period—could help identify individuals with persistently high viral loads. This would allow clinicians to maintain the current regimen for those who do suppress, while targeting interventions for those who do not. In our context, the alternative would likely be either delaying regimen switches even further or not switching at all.

  1. I would have expected that the authors would advocate for the ability to do in-country HIVDR testing for regular HIVDR surveillance and possibly for individual case management (vs. sending samples to another continent).

Response:

We did. In fact, this samples served (in parallel) as a training for the local institution in charge of the HIVDR surveillance, because it is no easy to find such a high number of INSTI mutations in the same cohort. But this is not the focus of our analysis, and we prefer not to mention it. Unfortunately the genotype access for clinical purposes is far from happening in the current financial context.

Specific minor points

  1. Line 71. “Mozambique's lessons offer critical insights for other sub-Saharan African countries navigating DTG transitions”. This is not a sentence that fits into a Background section, it would be expected in the Discussion, and the only of course if “critical lessons” can be learned from the findings.

Response:

We agree. Removed.

  1. Line 121. “Eligible participants included individuals of all ages who had access to HIV-GT.” I think the wording is not accurate. Participants were included if they were selected for HIVDR testing.

Response:

We agree: The sentence is change as follows: Participants selected for HIVDR testing were included.

  1. Line 158. “those who tested positive for resistance” – indicate if this is DTG resistance, or if not, clarify what kind of resistance is meant

Response:

It is offered as an example to define PPV. The example was completed as follows: those who tested positive for resistance to a certain drug.

  1. Line 207. “ART drug combination”. Is this the regimen that the individual was on when the sample for sequencing was taken? please clarify.

Response:

Yes, clarified as follows: ART drug combination at the time of the Genotype sample collection

  1. Line 246. “Subtype C 99 (93.4%)” This is a repeat. Please remove and in addition, please decide on the number of decimals for percentages and numbers and use this consistently throughout text, tables and figures

Response:

Done, thanks.

  1. Line 249. It is argued that men have a higher percentage of AHD than women but neither in the text, nor in Figure 2 are point estimates provided, which makes it hard to gauge the clinical relevance and magnitude of this finding. Please provide point estimates and numbers of the column totals and categories

Response: We absorbed

Table 1: Cohort main characteristics

All

DTG-based

PI-based

106 

62 

44 

Sex

Women, N (%) 

57 (53.8) 

30 (48.4) 

27 (61.4) 

Men, N (%)

49 (46.2)

35 (53.8)

17 (38.6)

  1. Line 277. “so we assume we didn´t find primary resistance”. This is a conclusion that should appear in the Discussion, not the Results section

Response:

We agree. It has been removed from line 249 and kept only in the discussion part (line 390)

  1. Line 278. PI resistance mutations were seen among individuals on a DTG regimen, had they been exposed to PIs in previous regimens? Such information would be valuable

Response:

Some of them had. To have the exact data we would need to review all files, as the information was not collected initially.

  1. Line 321. “resistance rate”. Given that percentages are presented, I would suggest saying ratio instead of rate

Response:

Corrected.

  1. Lines 355-374. This paragraph contains much repetition of results and the relevance of the findings is not made very clear. I suggest considerable shortening and providing meaning of the mutation pattern findings observed (if there is any).

Response:

Text simplified and repetition deleted.

Reviewer 2 Report (Previous Reviewer 3)

Comments and Suggestions for Authors

Dear Authors,

congratulations on your revised version of the paper, I really appreciate your efforts and I acknowledge that your work has sufficiently improved in quality.

Author Response

August 31, 2025

Infectious Disease reports (idr),

EBM - Editorial Board Member

  • Steve Chen
  • Marija Jovanović

Dear Editors:

Please find attached a revised version of our manuscript, “Dolutegravir resistance in Mozambique:  insights from a programmatic HIV resistance testing intervention in a highly antiretroviral therapy experienced cohort  ,” (Manuscript idr-3726835 or idr-3618516). We have modified the original submission taking into consideration all comments and suggestions made by you. All changes have been highlighted in tracked changes on revised document. Below, please find an itemized response.

Answer (A): We appreciate your thanks and, considering it extremely important for the enrichment of the article, we added, deleted, rewrote, and reorganized the recommended points, according to your suggestions throughout the document.

Reviewer 2

Congratulations on your revised version of the paper, I really appreciate your efforts and I acknowledge that your work has sufficiently improved in quality.

Response:

Thank you very much for your appreciation. We are working on it, absorbing your comments.

Reviewer 3 Report (Previous Reviewer 4)

Comments and Suggestions for Authors

The study addresses a critically important issue, the emerging dolutegravir (DTG) resistance, in a high-risk, resource-limited setting. The authors are commended for leveraging programmatic data to fill an urgent knowledge gap.

- The distinction between major vs. non-major DTG mutations is useful, but a clearer breakdown of how these categories impact clinical decision-making (e.g., when DTG can still be retained) would improve the discussion. Also, it is recommended to separate the majors from the minors in two figures.

- The gender-based disparities in immunologic status are briefly noted; however, more discussion on gender-specific adherence barriers or healthcare access would strengthen the public health relevance in the discussion part.

- Several grammar and syntax errors exist throughout the manuscript, which may impede readability.

- The study would benefit from a visual schematic/chart showing the resistance score of each mutation.

- The quality of the figures needs to be strengthened, especially for figure 1 and 2.

Comments on the Quality of English Language

Carefully check grammar and abbreviations. 

Author Response

August 31, 2025

Infectious Disease reports (idr),

EBM - Editorial Board Member

  • Steve Chen
  • Marija Jovanović

Dear Editors:

Please find attached a revised version of our manuscript, “Dolutegravir resistance in Mozambique:  insights from a programmatic HIV resistance testing intervention in a highly antiretroviral therapy experienced cohort  ,” (Manuscript idr-3726835 or idr-3618516). We have modified the original submission taking into consideration all comments and suggestions made by you. All changes have been highlighted in tracked changes on revised document. Below, please find an itemized response.

Answer (A): We appreciate your thanks and, considering it extremely important for the enrichment of the article, we added, deleted, rewrote, and reorganized the recommended points, according to your suggestions throughout the document.

Reviewer 3

The study addresses a critically important issue, the emerging dolutegravir (DTG) resistance, in a high-risk, resource-limited setting. The authors are commended for leveraging programmatic data to fill an urgent knowledge gap.

  1. The distinction between major vs. non-major DTG mutations is useful, but a clearer breakdown of how these categories impact clinical decision-making (e.g., when DTG can still be retained) would improve the discussion. Also, it is recommended to separate the majors from the minors in two figures.

Response:

We truly appreciate you raising this point about separating major and minor; it's an excellent thought. To ensure we're all aligned, our view is that keeping the figure as it stands will best serve our common understanding.

  1. The gender-based disparities in immunologic status are briefly noted; however, more discussion on gender-specific adherence barriers or healthcare access would strengthen the public health relevance in the discussion part.

Response:

We agree that differences are notable, but if we open this discussion, the focus of the article might change. We will address it in another publication.

  1. Several grammar and syntax errors exist throughout the manuscript, which may impede readability.

Response:

We absorbed

  1. The study would benefit from a visual schematic/chart showing the resistance score of each mutation.

Response:

We absorbed

  1. The quality of the figures needs to be strengthened, especially for figure 1 and 2.

Response:

We absorbed

Round 2

Reviewer 1 Report (Previous Reviewer 1)

Comments and Suggestions for Authors

1 Reference to the "relative DTG resistance score" still appears in the abstract. This must be removed

2 I would still recommend that the authors acknowledge that the the very high percentage of high-level resistance and the very high percentage of AHD argues for long delays in virological failure management and may have been contributed to by the diagnostic MSF-iTECH algorithm. A more active approach of viremia management may be recommended, for instance with HIVDR testing that is nationally available 

Author Response

Comments and Suggestions for Authors

  1. Reference to the "relative DTG resistance score" still appears in the abstract. This must be removed

Answer: We agreed, sentences removed

  1. 2 I would still recommend that the authors acknowledge that the the very high percentage of high-level resistance and the very high percentage of AHD argues for long delays in virological failure management and may have been contributed to by the diagnostic MSF-ITECH algorithm. A more active approach of viremia management may be recommended, for instance with HIVDR testing that is nationally available.

Answer – We agreed, the sentence is change as follows, see lines 336 – 345 on clean document.

It is plausible to consider that the elevated prevalence of antiretroviral resistance observed in this cohort may have been partially influenced by delays in case identification, potentially associated with the diagnostic algorithm employed. In clinical practice, this patient cohort remained in virological failure for a prolonged period prior to referral to our healthcare unit, which may have contributed to the accumulation of resistance mutations.

In healthcare settings with broader access to genotypic resistance testing, reliance on such algorithms might be less critical, as earlier detection could facilitate more timely therapeutic adjustments and potentially mitigate the development of resistance. Nevertheless, in our context

This manuscript is a resubmission of an earlier submission. The following is a list of the peer review reports and author responses from that submission.

Round 1

Reviewer 1 Report

Comments and Suggestions for Authors

Dolutegravir resistance in Mozambique: insights from a pro-grammatic HIV resistance testing intervention in a highly ART experienced cohort

Maria Ruano , Flores Flores , Aleny Couto , Irénio Gaspar , Sabine Yerly , Ana Gabriela Gutierrez Zamudio, Rosa Bene , Adelina Maiela , Helder Macuacua , Jeff Lane , Florindo Mudender , Edy Nacarpa

Journal: Infectious Disease Reports

Date 25 April 2025

This is a survey of genotypic HIVDR testing results from a selected population of experienced ART clients who experienced long-term virological failure, often complicated by advanced HIV disease (AHD) after a clinical algorithm selected them for HIVDR testing. The survey is from an MSF supported program in Mozambique and provides interesting insight into the current challenges of managing virological failure in individuals on dolutegravir and PI-based ART regimens in settings with limited diagnostic options, specifically a strong limitation in HIVDR testing.

Major points

  1. The study population is highly selected and not representative of individuals on ART, nor of individuals with virological failure as commonly defined. A bullet point at the end mentions: “High prevalence of DTG resistance”. There and elsewhere in the manuscript, the authors should avoid the term prevalence when referring to percentages of HIVDR and associated mutations observed, given the lack of representativeness of the study population
  2. The benefit of the MSF algorithm to select individuals for HIVDR testing needs to be toned down considerably in my view. First, this is not a study that demonstrated test characteristics or clinical benefit with relevant outcomes, so formal conclusions cannot be drawn. There was a high percentage of HIVDR among those tested, but we do not know anything about those who did not fulfil criteria of the algorithm: it seems likely that many with HIVDR were missed among these. Second, the algorithm requires a protracted and intensive process and has potential disadvantages, including a) delays in ART switching with immune deterioration as a possible consequence, b) accumulation of DTG resistance during continued viremia and drug exposure and c) loss to FU due to frequent (mandatory?) clinic visits that are costly for clients. The authors say “Our algorithm demonstrated a notably high positive predictive value to find resistance (over 89%), among individuals with significant exposure to ART”. Can the authors clarify how they determined PPV and what “significant” exposure to ART is? In the Discussion and in the bullet points at the end (“Effective Algorithm for Resistance Identification”) I strongly recommend to pay attention to these points and provide a much more reserved narrative on the algorithm.

Minor points

  1. The introduction should end with the aim or hypothesis of the study: why was this report written?
  2. “In 2003, Médecins Sans Frontières (MSF) established the Centro de Referência de Alto-Mae (CRAM) in collaboration with the Ministry of Health (MoH) as an outpatient center dedicated to the referral of PLWH with advanced HIV disease (AHD) and those suspected of experiencing failure with second-line ART regimens from the broader Maputo urban area”. Do the authors mean clients had failure on 2nd line ART regimens? Please clarify
  3. The authors may include that a “cross-sectional assessment of routine programmatic data” is a limitation
  4. “The sole exclusion criterion for eligibility to undergo a genotyping test was a viral load result of less than 5000 copies/ml after adherence support interventions. This limitation arises from the inherent constraints of dried blood spot (DBS) samples, which are not capable of effectively amplifying and sequencing viral genetic material at lower viral load levels”. Can the authors provide a reference for this statement? It is by no means impossible to successfully sequence from DBS samples with a VL <5000
  5. “Resistance scores for tenofovir (TDF), zidovudine (AZT), atazanavir (ATV) and dolutegravir (DTG) are reported. In addition, we present a composite class-related nucleotide reverse transcriptase inhibitor (NRTI) resistance score calculated by averaging the TDF-R and AZT-R scores (where one of the scores was not available, the NRTI-R score was imputed as the other available score) [12].” What is the meaning of a class resistance score and is this a conventional indicator? Note that this paragraph is copied-repeated but with different references in the repeat text!
  6. In Table 1, it would be valuable if the duration of viremia on a DTG regimen is included
  7. In Results: “and 51 (89.5% [95% CrI: 80.7, 96.2]) had confirmed resistance to DTG”. How was DTG resistance defined here?
  8. “The mean relative DTG-R score (the DTG-R score standardized by total number of INSTI mutations) was 20.6 (95% CrI: 19, 22.2). The relative DTG-R score was positively correlated with the number of major mutations (bootstrapped Pearson’s correlation coefficient, r: 0.61 [95% CrI: 0.42, 0.77; pd: 100%]).” Are “DTG-R scores” the same as ARV drug penalty scores in Stanford? The determination of a relative DTG-R score should be explained in the methods. It should also be indicated what the meaning and clinical relevance of this score is
  9. The resolution of Figure 2 is insufficient
  10. In Figure 3, add to the title: among individuals on DTG
  11. In the Discussion: “In this programmatic HIV-GT intervention, DTG-R was consistently found in highly ART exposed persons experiencing VF, after going through a structured adherence algorithm that included a short, supervised treatment intervention.” DTG resistance was not consistently found: only in clients on DTG and not in all of them.
  12. In the 2nd paragraph of the Discussion: which conclusions do the authors draw from the very high percentage of DTG resistance?
  13. “This mutation (E138K) is categorized as a non-major mutation, associated with low-level resistance to DTG when it occurs in isolation”. The Stanford website says about E138 mutations: alone they do not reduce INSTI susceptibility. However, they contribute to reduced susceptibility in combination with other mutations – how does this align with the authors’ statement? 
  14. “Furthermore, the third most common major mutation is E148K,”. Do the authors mean Q148K?
  15. When comparing resistance patterns of their study with others, the HIV subtype is relevant and the (dominant) HIV subtype in other studies needs to be mentioned, as the prevalence of specific mutations and of mutation patterns may vary between HIV subtypes
  16. “Consequently, we can conclude that primary resistance to integrase strand transfer inhibitors (INSTIs) was not identified, despite the extended duration of antiretroviral therapy (ART”. Please explain this sentence. How does primary (transmitted) resistance relate to the extended duration of ART?
  17. “(median composite score for NRTI resistance over 5)”, please explain what this means, it was not clear to me
  18. In the Bullet points, under “Programmatic implications”, the authors should consider the comparative need for HIVDR testing for individual management vs. the need for regular HIVDR surveillance, in settings where affordability and feasibility of HIVDR testing is challenging
  19. “Mutational Patterns and Resistance Scores……The correlation between the number of major mutations and resistance scores further elucidates the mechanisms of resistance development.” I believe this is overstated as the resistance scores (if these are the same as penalty scores at the Stanford website) are provided by the Stanford website based on algorithms and data in the Stanford software and this survey of a small number of sequences does not really add new insights. Or if they do, what exactly are these insights?

Author Response

This is a survey of genotypic HIVDR testing results from a selected population of experienced ART clients who experienced long-term virological failure, often complicated by advanced HIV disease (AHD) after a clinical algorithm selected them for HIVDR testing. The survey is from an MSF supported program in Mozambique and provides interesting insight into the current challenges of managing virological failure in individuals on dolutegravir and PI-based ART regimens in settings with limited diagnostic options, specifically a strong limitation in HIVDR testing.

Major points

  1. The study population is highly selected and not representative of individuals on ART, nor of individuals with virological failure as commonly defined. A bullet point at the end mentions: “High prevalence of DTG resistance”. There and elsewhere in the manuscript, the authors should avoid the term prevalence when referring to percentages of HIVDR and associated mutations observed, given the lack of representativeness of the study population

Response: Thank you for pointing this out. We agree with this comment. Therefore, we have accordingly, revised the previous statement to emphasize this point. “see line 455, 456”

Done: The word prevalence has been replaced by the word detection in 2 places

  1. The benefit of the MSF algorithm to select individuals for HIVDR testing needs to be toned down considerably in my view. First, this is not a study that demonstrated test characteristics or clinical benefit with relevant outcomes, so formal conclusions cannot be drawn. There was a high percentage of HIVDR among those tested, but we do not know anything about those who did not fulfil criteria of the algorithm: it seems likely that many with HIVDR were missed among these.

Response: Thank you for pointing this out. We agree with this comment. Therefore, we have accordingly, revised the previous statement to emphasize this point. “see lines 137 – 139: The sole exclusion criterion for eligibility to undergo a genotyping resistance test was a viral load result of less than 5000 copies/ml after adherence support interventions, as per the laboratory standard operating practice.”

Unfortunately, we are not presenting here data from patients that entered the flow and didn’t end up getting a genotype test, but most of them were removed from the flow because VL suppression was reached. Some other reduced considerably the VL levels and because of falling under 5000 c/ml threshold, didn´t have access to Geno test.

  1. Second, the algorithm requires a protracted and intensive process and has potential disadvantages, including a) delays in ART switching with immune deterioration as a possible consequence, b) accumulation of DTG resistance during continued viremia and drug exposure and c) loss to FU due to frequent (mandatory?) clinic visits that are costly for clients. The authors say, “Our algorithm demonstrated a notably high positive predictive value to find resistance (over 89%), among individuals with significant exposure to ART”. Can the authors clarify how they determined PPV and what “significant” exposure to ART is? In the Discussion and in the bullet points at the end (“Effective Algorithm for Resistance Identification”) I strongly recommend paying attention to these points and provide a much more reserved narrative on the algorithm.

Response: Thank you for pointing this out. We agree with this comment. Therefore, we have accordingly, revised the previous statement to emphasize this point. “See lines 161 – 164: The positive predictive value (PPV) is defined as the proportion of individuals with the target condition (i.e., those who tested positive for resistance) among all those who tested positive according to the diagnostic algorithm. Mathematically, PPV is calculated as the ratio of true positives (TP) to the sum of true positives and false positives (FP).” and “see lines 348 – 355:  In this context, the algorithm classified individuals as positive if they did not achieve virological suppression following enhanced adherence support. The analysis demonstrated that approximately 89% of those identified as "algorithm-positive" (i.e., non-suppressors despite adherence interventions) had a confirmed positive resistance test, irrespective of the resistance level. Thus, the PPV of the algorithm for predicting a positive resistance test was 89%, indicating that an algorithm-based positive result correctly anticipated detectable resistance in 89% of cases.”

I agree with the disadvantages listed regarding this algorithm. It might take longer time (6 moths) to select and give access to genotype to patients. The problem is that access to this kind of test is scarce in the context. The alternative is not having access to it. I will read carefully the text again, to make sure that we are not “hiding” the disadvantages.

We define the PPV as the capacity of a certain test (in this case, our algorithm) to detect resistance to DTG. We are saying that the PPV os the algorithm was the positivity of resistance mutations encountered, divided by the total number of geno tests performed.

“Significant exposure to ART” is used to avoid the generalization of the conclusion. The algorithm probably performs differently in groups with lower resistance presence.

Minor points

  1. The introduction should end with the aim or hypothesis of the study: why was this report written?

Response: Thank you for pointing this out. We agree with this comment. Therefore, we have accordingly, revised the previous statement to emphasize this point. “ see lines 89 – 91: The aim of this analysis is to share our field experience in identifying cost-effective strategies to detect ART resistance in a resource-limited setting.”

Done: added a sentence at the introduction with the aim of the analysis

  1. “In 2003, Médecins Sans Frontières (MSF) established the Centro de Referência de Alto-Mae (CRAM) in collaboration with the Ministry of Health (MoH) as an outpatient center dedicated to the referral of PLWH with advanced HIV disease (AHD) and those suspected of experiencing failure with second-line ART regimens from the broader Maputo urban area”. Do the authors mean clients had failure on 2nd line ART regimens? Please clarify

Response: Thank you for pointing this out. We agree with this comment. Therefore, we have accordingly, revised the previous statement to emphasize this point. “see lines 108 – 112: In 2003, Médecins Sans Frontières (MSF) established the Centro de Referência de Alto-Mae (CRAM) in collaboration with the Ministry of Health (MoH) as an outpatient center dedicated to the referral of PLWH with advanced HIV disease (AHD) and those suspected of ART failure and in need of second-line ART regimens, from the broader Maputo urban area.”

CRAM was established to support the management of patients that were difficult to handle in the primary level HF in Maputo city. That included AHD patients, but also patients with persistent high VL in use of first line ART (there was a mistake in the text, please see correction in the article). During the first decade of ART in country, CRAM centralized patients that needed a switch to 2nd line (that means failing first line). After 2013, 2nd line treatment was decentralized to all HF and most of those patients were sent back to other HF.

  1. The authors may include that a “cross-sectional assessment of routine programmatic data” is a limitation

Response: Thank you for pointing this out. We agree with this comment. Therefore, we have accordingly, revised the previous statement to emphasize this point. “See lines 431 – 433: Additionally, the cross-sectional nature of this analysis did not allow for the evaluation of ART management (regimen switch) based on resistance test results, limiting the understanding of the impact of HIV resistance testing on longer-term out-comes.”

  1. “The sole exclusion criterion for eligibility to undergo a genotyping test was a viral load result of less than 5000 copies/ml after adherence support interventions. This limitation arises from the inherent constraints of dried blood spot (DBS) samples, which are not capable of effectively amplifying and sequencing viral genetic material at lower viral load levels”. Can the authors provide a reference for this statement? It is by no means impossible to successfully sequence from DBS samples with a VL <5000

Response: Thank you for pointing this out. We agree with this comment. Therefore, we have accordingly, revised the previous statement to emphasize this point. “See lines 137 – 141: The sole exclusion criterion for eligibility to undergo a resistance test was a viral load result of less than 5000 copies/ml after adherence support interventions, as per the laboratory standard operating practice. This limitation arises from the inherent constraints of dried blood spot (DBS) samples, which are not capable of effectively amplifying and sequencing viral genetic material at lower viral load levels [10].”

Reference: WHO manual for HIV drug resistance testing using dried blood spot specimens, third edition. Geneva: World Health Organization; 2020. Licence: CC BY-NC-SA 3.0 IGO

In page 3 (table) you can find the amplification success rates and some of the studies show how performance goes down with the VL level.

  1. “Resistance scores for tenofovir (TDF), zidovudine (AZT), atazanavir (ATV) and dolutegravir (DTG) are reported. In addition, we present a composite class-related nucleotide reverse transcriptase inhibitor (NRTI) resistance score calculated by averaging the TDF-R and AZT-R scores (where one of the scores was not available, the NRTI-R score was imputed as the other available score) [12].” What is the meaning of a class resistance score and is this a conventional indicator? Note that this paragraph is copied-repeated but with different references in the repeat text!

Response: Thank you for pointing this out. We agree with this comment. Therefore, we have accordingly, revised the previous statement to emphasize this point. “See lines: 179 – 187: Resistance (of any level: from potential low level to high level) was defined as a penalty score ≥ 5. To account for differences in the total number of INSTI resistance mutations observed across isolates, we calculated a relative DTG-R score by standardizing the raw DTG-R score (i.e., the sum of weighted mutation penalties) by the total number of INSTI mutations in each sequence. This normalization adjusts for variability in mutation burden, allowing for more comparable resistance estimates between viral strains with differing numbers of mutations.  A higher relative DTG-R indicates a greater per-mutation resistance impact, suggesting that the observed mutations collectively confer stronger resistance to dolutegravir (DTG) independent of mutation quantity.”

  1. In Table 1, it would be valuable if the duration of viremia on a DTG regimen is included

Response: Thank you for pointing this out. We agree with this comment. Therefore, we have accordingly, revised the previous statement to emphasize this point. “See lines 339 – 346: As this study relied on real-world programmatic data, detailed historical viral load testing records were often incomplete or unavailable for analysis. However, available programmatic evidence indicates substantial delays in initial high viral load detection, frequently exceeding 6 months, due to gaps in routine monitoring and reporting. When combined with the standard 4-month enhanced adherence counselling (EAC) process prior to resistance testing, our conservative estimate suggests that participants in the pre-resistance testing ART support algorithm experienced at least 11 months of viremia before regimen adjustment.”

  1. In Results: “and 51 (89.5% [95% CrI: 80.7, 96.2]) had confirmed resistance to DTG”. How was DTG resistance defined here?

Response: Thank you for pointing this out. We agree with this comment. Therefore, we have accordingly, revised the previous statement to emphasize this point. “See 179 – 187: Resistance (of any level: from potential low level to high level) was defined as a penalty score ≥ 5. To account for differences in the total number of INSTI resistance mutations observed across isolates, we calculated a relative DTG-R score by standardizing the raw DTG-R score (i.e., the sum of weighted mutation penalties) by the total number of INSTI mutations in each sequence. This normalization adjusts for variability in mutation burden, allowing for more comparable resistance estimates between viral strains with differing numbers of mutations.  A higher relative DTG-R indicates a greater per-mutation resistance impact, suggesting that the observed mutations collectively confer stronger resistance to dolutegravir (DTG) independent of mutation quantity.”

  1. “The mean relative DTG-R score (the DTG-R score standardized by total number of INSTI mutations) was 20.6 (95% CrI: 19, 22.2). The relative DTG-R score was positively correlated with the number of major mutations (bootstrapped Pearson’s correlation coefficient, r: 0.61 [95% CrI: 0.42, 0.77; pd: 100%]).” Are “DTG-R scores” the same as ARV drug penalty scores in Stanford? The determination of a relative DTG-R score should be explained in the methods. It should also be indicated what the meaning and clinical relevance of this score is

Response: Thank you for pointing this out. We agree with this comment. Therefore, we have accordingly, revised the previous statement to emphasize this point.

  • “See lines 179 – 187: Resistance (of any level: from potential low level to high level) was defined as a penalty score ≥ 5. To account for differences in the total number of INSTI resistance mutations observed across isolates, we calculated a relative DTG-R score by standardizing the raw DTG-R score (i.e., the sum of weighted mutation penalties) by the total number of INSTI mutations in each sequence. This normalization adjusts for variability in mutation burden, allowing for more comparable resistance estimates between viral strains with differing numbers of mutations.  A higher relative DTG-R indicates a greater per-mutationresistance impact, suggesting that the observed mutations collectively confer stronger resistance to dolutegravir (DTG) independent of mutation quantity.”
  • “See lines 393 – 401: We calculated a relative DTG-R score, defined as the DTG-R score divided by the total number of observed INSTI-associated resistance mutations in each sequence. The relative DTG-R score provides insight into the qualitative potency of resistance mutations rather than their sheer number. This standardized metric provides an estimate of the proportion of INSTI mutations contributing specifically to dolutegravir resistance, rather than resistance to the broader drug class. The relative DTG-R score may help distinguish profiles where dolutegravir resistance is a dominant feature from those where mutations are more diffusely distributed across multiple INSTIs. Clinically, this may assist in identifying cases where dolutegravir is likely to remain effective despite the presence of INSTI mutations.”

  1. The resolution of Figure 2 is insufficient

Response: Thank you for pointing this out. We agree with this comment. Therefore, we have accordingly, revised the previous statement to emphasize this point. “See new figure 2 adjusted”

  1. In Figure 3, add to the title: among individuals on DTG

Response: Thank you for pointing this out. We agree with this comment. Therefore, we have accordingly, revised the previous statement to emphasize this point. “See line 306 – 307: Frequency of DTG mutations among individuals on DTG and with DTG resistance mutations.

  1. In the Discussion: “In this programmatic HIV-GT intervention, DTG-R was consistently found in highly ART exposed persons experiencing VF, after going through a structured adherence algorithm that included a short, supervised treatment intervention.” DTG resistance was not consistently found: only in clients on DTG and not in all of them.

Response: Thank you for pointing this out. We agree with this comment. Therefore, we have accordingly, revised the previous statement to emphasize this point. “See Lines 320 – 323: In this programmatic HIV-GT intervention, DTG-R was very frequently found in highly ART exposed persons on a DTG-based ART regimen experiencing VF, after go-ing through a structured adherence algorithm that included a short, supervised treat-ment intervention.”

  1. In the 2nd paragraph of the Discussion: which conclusions do the authors draw from the very high percentage of DTG resistance?

Response: Thank you for pointing this out. We agree with this comment. Therefore, we have accordingly, revised the previous statement to emphasize this point. “See lines 324 – 326: In our cohort, 51 cases of DTG resistance were detected among 57 selected PLWH resulting in a resistance rate of 89.5% in individuals experiencing confirmed virological failure. This finding indicates a high resistance detection within our cohort.”

I would say that the conclusions are explained in paragraphs 3 and 4 of the discussion: The high yield of resistance found is since we filter patients very conscientiously with a high-quality adherence support strategy, followed by a supervised treatment period of 4 weeks. Only patients that persist without VL suppression are tested with genotype. The conclusion is that these high-quality adherence support strategies are necessary and would allow for targeted allocation of resistance testing.

  1. “This mutation (E138K) is categorized as a non-major mutation, associated with low-level resistance to DTG when it occurs in isolation”. The Stanford website says about E138 mutations: alone they do not reduce INSTI susceptibility. However, they contribute to reduced susceptibility in combination with other mutations – how does this align with the authors’ statement? 

Response: Thank you for pointing this out. We agree with this comment. Therefore, we have accordingly, revised the previous statement to emphasize this point. “See lines 268 – 270: However, the most frequent mutation found was E138K (n=33), a non-major mutation that only confers potential low-level resistance when present isolated.”

  1. “Furthermore, the third most common major mutation is E148K,”. Do the authors mean Q148K?

Response: Thank you for pointing this out. We agree with this comment. Therefore, we have accordingly, revised the previous statement to emphasize this point. “See line 364 – 365: the third most common major mutation was Q148K, identified in 7 (14%) out of 51 samples.”

  1. When comparing resistance patterns of their study with others, the HIV subtype is relevant and the (dominant) HIV subtype in other studies needs to be mentioned, as the prevalence of specific mutations and of mutation patterns may vary between HIV subtypes

Response: Thank you for pointing this out. We agree with this comment. Therefore, we have accordingly, revised the previous statement to emphasize this point. “See lines 356 – 369: When comparing various mutational patterns identified in previous cohort analyses, our study confirmed a Subtype C predominance and its of resistance mutations distribution across Subtypes (A, B, C and U).  This variation aligns with known differences in mutational patterns varies between HIV subtypes (A, C, D, URFs) in Southern and East Africa, a region dominated by subtype C.”

  1. “Consequently, we can conclude that primary resistance to integrase strand transfer inhibitors (INSTIs) was not identified, despite the extended duration of antiretroviral therapy (ART”. Please explain this sentence. How does primary (transmitted) resistance relate to the extended duration of ART?

Response: Thank you for pointing this out. We agree with this comment. Therefore, we have accordingly, revised the previous statement to emphasize this point. “See lines 386 – 390: In the cohort of PLWH on a protease inhibitor-based regimen, there was no recorded prior exposure to dolutegravir (DTG), and primary resistance to integrase strand transfer inhibitors (INSTIs) was not identified, due to the extended duration of antiretroviral therapy (ART) dating back to a pre-DTG period”

What we try to say is that primary resistance is not found to INSTI. If the patient has not been exposed to INSTI, no INSTI mutations are found. We could have found mutations that were present (ex, not necessarily the major mutations, but maybe others), but it is not the case.

  1. “(median composite score for NRTI resistance over 5)”, please explain what this means, it was not clear to me

Response: Thank you for pointing this out. We agree with this comment. Therefore, we have accordingly, revised the previous statement to emphasize this point. “See lines 391: removed statement…”

“Reviewed sentences, see line 402 – 402: Extended resistance to other ART classes was observed, showing a high detection  of NRTI resistance in the cohort (overall, 76.5% for TDF, and 71.5% for AZT), including in the DTG subgroup.”

  1. In the Bullet points, under “Programmatic implications”, the authors should consider the comparative need for HIVDR testing for individual management vs. the need for regular HIVDR surveillance, in settings where affordability and feasibility of HIVDR testing is challenging.

Response: Thank you for pointing this out. We agree with this comment. Therefore, we have accordingly, revised the previous statement to emphasize this point. “See lines 475 – 478: It also highlights the importance of elucidating the role of regular HIVDR surveillance vs clinically driven individual resistance testing in settings where affordability and feasibility of HIVDR testing are perceived as limited.”

  1. “Mutational Patterns and Resistance Scores……The correlation between the number of major mutations and resistance scores further elucidates the mechanisms of resistance development.” I believe this is overstated as the resistance scores (if these are the same as penalty scores at the Stanford website) are provided by the Stanford website based on algorithms and data in the Stanford software and this survey of a small number of sequences does not really add new insights. Or if they do, what exactly are these insights?

Response: Thank you for pointing this out. We agree with this comment. Therefore, we have accordingly, revised the previous statement to emphasize this point. “ sentences removed”

Reviewer 2 Report

Comments and Suggestions for Authors
  1. Define PLWH in abstract and introduction.
  2. Introduction: It would be helpful to provide more information on DTG resistance management.
  3. Methodology: Some methods are suggested to be more detailed. For example, the detection method of CD4.
  4. Table 1 and 2: Have authors compared the differences of parameters between DTG-based and PI-based?
  5. The figure legends of Figure 2 and 3 should be more detailed.
  6. The Results section lacks a clear structure, which makes it difficult for readers to read. It would be helpful to reorganize this section.
  7. Proofreading is suggested.

Author Response

  1. Define PLWH in abstract and introduction.

Response: Thank you for pointing this out. We agree with this comment. Therefore, we have accordingly, revised the previous statement to emphasize this point. “See lines 31 – 32: People living with HIV (PLWH)”

  1. Introduction: It would be helpful to provide more information on DTG resistance management.

Response: Thank you for pointing this out. We agree with this comment. Therefore, we have accordingly, revised the previous statement to emphasize this point. “See lines 92 – 98: This evaluation aimed to assess the ability to detect resistance within our patient flow. Patient management is individualized based on genotype test results. Most cases exhibiting Dolutegravir (DTG) resistance were offered salvage therapy consisting of a boosted protease inhibitor (PI) and two nucleoside reverse transcriptase inhibitors (NRTIs). However, in specific instances where DTG retained partial activity – particularly when extensive resistance to NRTIs was present – DTG was maintained within the treatment regimen”

  1. Methodology: Some methods are suggested to be more detailed. For example, the detection method of CD4.

Response: Thank you for pointing this out. We agree with this comment. Therefore, we have accordingly, revised the previous statement to emphasize this point. “See lines 216 – 217: (6) most recent CD4 T lymphocyte count (measured using PIMA technology).”

  1. Table 1 and 2: Have authors compared the differences of parameters between DTG-based and PI-based?

Response: Thank you for pointing this out. We agree with this comment. Therefore, we have accordingly, revised the previous statement to emphasize this point.

We didn´t find significant differences, except for time of exposure to the current ART regime, slightly longer for patients on IP. This comparison was not the focus of the analysis, but still it can be compared in the tables.

  1. The figure legends of Figure 2 and 3 should be more detailed.

Response: Thank you for pointing this out. We agree with this comment. Therefore, we have accordingly, revised the previous statement to emphasize this point.

  • “See lines 299 – 302: Legend: Figure 2 revealed a difference in immunological status based on gender among people living with HIV (PLHIV) in this cohort. Male participants demonstrated a lower mean CD4 count, and a higher rate of advanced HIV disease compared to females. These differences had a higher probability, with a prevalence difference of 24.6% (95% CrI: 2.2-46).”
  • “See lines 309 – 313: Legend: Figure 3 showed that major dolutegravir (DTG)-associated resistance mutations were detected in 46 of 57 samples (80.7%). The specific major mutations identified were G118R (28 samples), R263K (15 samples), and Q148R/K (7 samples). However, the most frequently observed mutation overall was E138K (33 samples), which is not classified as a major DTG mutation and typically confers only low-level resistance when it occurs in isolation.”

  1. The Results section lacks a clear structure, which makes it difficult for readers to read. It would be helpful to reorganize this section.

Response: Thank you for pointing this out. We agree with this comment. Therefore, we have accordingly, revised the previous statement to emphasize this point. “See lines 246: …Cohort main characteristics” and “see line 262: …INSTI mutations and resistance.”

  1. Proofreading is suggested.

Response: Thank you for pointing this out. We agree with this comment. Therefore, we have accordingly, revised the previous statement to emphasize this point. “Done in whole document”

Reviewer 3 Report

Comments and Suggestions for Authors

Dear authors,

congratulations on your valuable work. Please, find here below some suggestions in order to further improve the quality of your paper.

  1. Please, avoid acronym in the title (or make it explicit).
  2. I think it would be beneficial for readers to have an expanded introduction. Aspects such as the safety implications related to dolutegravir use and compliance to integrase inhibitors therapy, should be mentioned. If you need some references, please take a look to these examples: https://pubmed.ncbi.nlm.nih.gov/32351747/; https://pubmed.ncbi.nlm.nih.gov/37632082/; https://pubmed.ncbi.nlm.nih.gov/37112904/; https://pubmed.ncbi.nlm.nih.gov/37549681/; https://pubmed.ncbi.nlm.nih.gov/38567806/
  3. Once again, beware of acronyms: they should be explicited the first time you mention it. Please, revise the whole paper accordingly.
  4. Figure 1 quality is very poor making it difficult to read, please improve the quality.
  5. Please, remove patents' section.

Author Response

  1. Please, avoid acronym in the title (or make it explicit).

Response: Thank you for pointing this out. We agree with this comment. Therefore, we have accordingly, revised the previous statement to emphasize this point. “See lines 2 – 4: Dolutegravir resistance in Mozambique:  insights from a programmatic HIV resistance testing intervention in a highly antiretroviral therapy experienced cohort

Done (ART = Antirretroviral)

  1. I think it would be beneficial for readers to have an expanded introduction. Aspects such as the safety implications related to dolutegravir use and compliance to integrase inhibitors therapy, should be mentioned. If you need some references, please take a look to these examples: https://pubmed.ncbi.nlm.nih.gov/32351747/; https://pubmed.ncbi.nlm.nih.gov/37632082/; https://pubmed.ncbi.nlm.nih.gov/37112904/; https://pubmed.ncbi.nlm.nih.gov/37549681/; https://pubmed.ncbi.nlm.nih.gov/38567806/
  2.  

Response: Thank you for pointing this out. We agree with this comment. Therefore, we have accordingly, revised the previous statement to emphasize this point. “See lines 69 – 77: Dolutegravir remains a transformative tool in Mozambique's HIV response, ena-bling 95% viral suppression rates in optimized programs. However, its safety advantages are compromised by systemic gaps in adherence support and resistance monitoring [3]. Urgent scale-up of viral load testing, targeted adherence interventions for high-risk groups, and sentinel resistance surveillance are essential to prevent widespread resistance [4]. The experience underscores that even high-barrier drugs like DTG are vulnerable to suboptimal implementation, particularly in regions dominated by HIV subtype C [5]. Mozambique's lessons offer critical insights for other sub-Saharan African countries navigating DTG transitions.”

  1. Once again, beware of acronyms: they should be explicated the first time you mention it. Please, revise the whole paper accordingly.

Response: Thank you for pointing this out. We agree with this comment. Therefore, we have accordingly, revised the previous statement to emphasize this point. “Done on whole document”

  1. Figure 1 quality is very poor making it difficult to read, please improve the quality.

Response: Thank you for pointing this out. We agree with this comment. Therefore, we have accordingly, revised the previous statement to emphasize this point. “Figure 1 revised”

  1. Please, remove patents' section.

Response: Thank you for pointing this out. We agree with this comment. Therefore, we have accordingly, revised the previous statement to emphasize this point. “There are on template document”

Reviewer 4 Report

Comments and Suggestions for Authors

The work has strong public health relevance for the use of DTG,

However, these points need to be addressed,

Major

-Study Design: The study is retrospective and cross-sectional, limited to individuals already showing treatment failure and high viral load. There is no comparator group of individuals on DTG without virologic failure or with lower ART exposure. This restricts the broader applicability of the algorithm and inflates the observed resistance rate.

-Data Analysis and Statistics: Bayesian bootstrapping is described in detail, but its advantage over frequentist methods is not well presented in the results. Some effect sizes (e.g., correlations) could be cross-validated with standard confidence intervals for transparency.

- Presentation and Clarity: Figures lack clarity, especially Figure 3 (mutation frequency), which should present the relative frequency of all mutations, not just a few highlighted ones.

- Limitations: Some limitations are acknowledged (e.g., use of dried blood spots), but the lack of phenotypic validation and real-world clinical outcomes (e.g., response to regimen switches) are not discussed.

Minor

-All figure quality needs to be improved e.g. 300 dpi.

Author Response

Major

-Study Design: The study is retrospective and cross-sectional, limited to individuals already showing treatment failure and high viral load. There is no comparator group of individuals on DTG without virologic failure or with lower ART exposure. This restricts the broader applicability of the algorithm and inflates the observed resistance rate.

Response: We agree. We make it clear that these results are not applicable to the general HIV population, and we make it clear that our cohort is a very particular one (highly ART experienced), even in the title of the article. Nevertheless, we think (and suggest) that this (or a similar) algorithm should be tested/validated in other groups.

-Data Analysis and Statistics: Bayesian bootstrapping is described in detail, but its advantage over frequentist methods is not well presented in the results. Some effect sizes (e.g., correlations) could be cross-validated with standard confidence intervals for transparency.

Response: We thank the reviewer for their thoughtful suggestion. We agree that cross-validating effect sizes with standard confidence intervals could enhance transparency through conventional interpretability.

Our choice of Bayesian nonparametric bootstrapping was motivated by the desire to avoid strong parametric and asymptotic assumptions (e.g., normality and large-sample approximations), which may not hold in our relatively small sample reflective of real-world programmatic conditions. Bayesian approaches, in this context, allow us to directly and probabilistically quantify uncertainty without relying on asymptotic normality.

That said, our main findings are not sensitive to the statistical framework used. In an additional analysis (available on request), we found that the Bayesian credible intervals align with standard frequentist confidence intervals, supporting the robustness of our findings. We are happy to include these comparisons in supplementary material if the reviewer or editors deem it valuable and fundamental to the manuscript.

- Presentation and Clarity: Figures lack clarity, especially Figure 3 (mutation frequency), which should present the relative frequency of all mutations, not just a few highlighted ones.

Response: Thank you for pointing this out. We agree with this comment. Therefore, we have accordingly, revised the previous statement to emphasize this point. “See lines 309 – 313: Legend: Figure 3 showed that major dolutegravir (DTG)-associated resistance mutations were detected in 46 of 57 samples (80.7%). The specific major mutations identified were G118R (28 samples), R263K (15 samples), and Q148R/K (7 samples). However, the most frequently observed mutation overall was E138K (33 samples), which is not classified as a major DTG mutation and typically confers only low-level resistance when it occurs in isolation.”

- Limitations: Some limitations are acknowledged (e.g., use of dried blood spots), but the lack of phenotypic validation and real-world clinical outcomes (e.g., response to regimen switches) are not discussed.

Response: Thank you for pointing this out. We agree with this comment. Therefore, we have accordingly, revised the previous statement to emphasize this point. “See lines 433 – 434: Lastly, discussion highlighting the focus on the identification algorithm, rather than on full resistance management.”

Yes, but the focus of the analysis was the performance of the screening system (the algorithm).

Minor

-All figure quality needs to be improved e.g. 300 dpi.

Response: Thank you for pointing this out. We agree with this comment. Therefore, we have accordingly, revised the previous statement to emphasize this point. “all figures reviewed”